# Genome-Wide Analysis of the *RbcS* Gene Family and Expression Analysis Under Light Response in *Brassica napus* L.

**DOI:** 10.3390/plants15010058

**Published:** 2025-12-24

**Authors:** Yanling Li, Cheng Cui, Liang Chai, Benchuan Zheng, Ka Zhang, Jun Jiang, Jinfang Zhang, Jing Wu, Jing Lang, Tongyun Zhang, Yongchun Zhou, Ping He, Liangcai Jiang, Hanzhong Wang, Haojie Li

**Affiliations:** 1Crop Research Institute, Sichuan Academy of Agricultural Sciences, Chengdu 610066, China; 2Key Laboratory of Biology and Genetic Improvement of Oil Crops, Ministry of Agriculture and Rural Affairs, Oil Crops Research Institute of the Chinese Academy of Agricultural Sciences, Wuhan 430062, China; 3Crop Germplasm Innovation and Genetic Improvement Key Laboratory of Sichuan Province, Chengdu 610066, China; 4Key Laboratory of Tianfu Seed Industry Innovation (Co-Construction by Ministry and Province), Ministry of Agriculture and Rural Affairs, Chengdu 610066, China

**Keywords:** *Brassica napus* L., *RbcS*, gene expression, light response, haplotype

## Abstract

Enhancing photosynthetic efficiency represents a key approach for improving crop biomass, with its translation into higher grain yield being contingent upon the efficiency of photosynthate partitioning toward harvestable organs. The Rubisco small subunit (*RbcS*) gene family plays an essential role in this process by stabilizing and regulating Rubisco assembly and activity during photosynthesis. In this study, we identified 61 *RbcS* genes across *B*. *napus*, *B*. *juncea*, and *B*. *carinata*, and their diploid progenitors *B*. *rapa*, *B*. *nigra*, and *B*. *oleracea* by genome-wide screening and bioinformatic approaches. Phylogenetic relationships, gene structures, conserved domains, collinearity, cis-regulatory elements, expression profiles, and haplotype variations were systematically investigated, revealing the potential functional role significance and regulatory complexity of *RbcS* genes in photosynthesis. The results imply that the promoter type of this gene family may belong to light-inducible promoters. Furthermore, while a haplotype analysis provided valuable insights for selecting germplasm with potentially high photosynthetic efficiency, definitive confirmation of their effects requires functional validation. Collectively, our results establish a theoretical foundation for understanding the molecular mechanisms of *BnRbcS* genes and propose candidate genetic targets for further exploration to enhance photosynthetic performance in rapeseed breeding.

## 1. Introduction

Rapeseed (*Brassica napus* L.) is the most important oil crop in China, and its yield and quality directly determine the stability and security of the national vegetable oil supply [1,2]. Against the backdrop of limited arable land resources, ongoing global climate change, and fluctuating import policies for vegetable oil, improving the productivity of oil crops has emerged as a critical component of national strategic security in China [3]. Photosynthesis constitutes the physiological basis for crop yield formation, contributing to 90–95% of biomass accumulation in crops. This process not only converts light energy into chemical energy but also provides the material foundation for agricultural production, with photosynthetic intensity directly affecting the final yield of crops [4,5,6]. Therefore, gaining a comprehensive and in-depth understanding of the evolutionary patterns and potential molecular functions of key genes involved in the photosynthetic pathway is of great significance for subsequent high-efficiency crop breeding via molecular design approaches [7,8].

In recent years, significant advancements have been achieved in understanding photosynthesis in rapeseed, particularly the physiological and molecular strategies it employs to optimize photosynthetic efficiency within the inherent constraints posed by abiotic stresses (e.g., salinity, shading, and nutrient limitation), as well as the potential of genetic interventions to alleviate these limitations and enhance photosynthetic capacity [9,10,11,12]. Specifically, leaf morphological traits—including leaf area, petiole angle, and chlorophyll content—play a pivotal role in optimizing light capture and carbon assimilation, significantly influencing photosynthetic performance, thereby directly regulating photosynthetic performance [13]. Beyond leaf traits, photosynthesis (a fundamental metabolic process in plants) is intricately modulated by core physiological factors, among which stomatal conductance and photosynthetic rate are paramount. These factors are closely linked to the catalytic efficiency of ribulose-1,5-bisphosphate carboxylase/oxygenase (Rubisco): the key enzyme mediating the first critical step of carbon fixation in the Calvin cycle. Clarifying the interplay between these regulatory elements is therefore crucial for improving crop productivity and environmental resilience [14,15]. Furthermore, the functional link between photosynthesis and oil accumulation in rapeseed seeds has recently attracted considerable attention. Given that photosynthetic carbon assimilation provides the primary carbon source for lipid biosynthesis, the functional link between photosynthetic efficiency and oil accumulation in rapeseed seeds has become a key focus for optimizing yield.

Ribulose-1,5-bisphosphate carboxylase/oxygenase (Rubisco) is a pivotal enzyme catalyzing atmospheric CO_2_ fixation in plant photosynthesis [16,17]. However, its catalytic efficiency is fundamentally limited by a slow carboxylation rate and poor specificity, allowing competitive oxygenation by O_2_ that leads to wasteful photorespiration [18]. Critically, the balance between carboxylation and oxygenation is modulated by key physiological and environmental factors: (1) the CO_2_/O_2_ ratio at the active site; (2) temperature, which affects CO_2_ solubility and enzyme stability; and (3) the alkaline pH (~8) of the chloroplast stroma, which is essential for Rubisco activation via Mg^2+^ binding [19,20]. These factors collectively constrain Rubisco’s in vivo efficiency. These inefficiencies have spurred extensive research into optimizing Rubisco’s catalytic properties and its regulation to enhance photosynthetic efficiency and crop productivity [21].

The catalytic efficiency of Rubisco is hindered by its bifunctionality, which not only catalyzes CO_2_ fixation but also competes with O_2_, leading to the phenomenon of photorespiration. Critically, the outcome of this competition is governed by the relative concentrations of CO_2_ and O_2_. Consequently, photorespiration reduces carbon assimilation efficiency, though plants have evolved mechanisms to mitigate its impact [22,23]. Rubisco is composed of a large subunit (RbcL) and a small subunit (RbcS). While RbcL encoded by the chloroplast genome contains the active site, the nuclear-encoded RbcS is essential for stabilizing the holoenzyme structure and regulating its assembly and activity [19]. The large subunit of RuBPCase is encoded by chloroplast genes and primarily functions catalytically, making it a valuable tool in the phylogenetic studies of plants to discern differences among biological species. In contrast, the small subunit is encoded by nuclear genes, which play a critical role in regulating RuBPCase activity. Ultimately, the assembly regulation of the holoenzyme is contingent upon the expression regulation of the *RbcS* gene [24]. The small subunit plays a significant role in the assembly and function of Rubisco, although it does not directly participate in the catalytic reaction. Through its interaction with RbcL, RbcS can influence the enzyme’s stability and catalytic efficiency [25,26]. In contrast to the relative conservation of the large subunit, the sequence of the RuBisCO small subunit exhibits greater diversity, with its common core structure composed of four antiparallel β-sheets, covered on one side by two α-helices [27].

The research indicates that the *RbcS* gene family exhibits considerable diversity among plant species, including copy number variation and divergence in regulatory sequences, which influence its expression patterns and stress responses and adaptive evolution to different environments. This diversity likely reflects adaptive evolution to different environments. In *Arabidopsis thaliana* (*A. thaliana*), four *AtRbcSs* (*RbcS*-*1A*, *RbcS*-*1B*, *RbcS*-*2B*, *RbcS*-*3B*) are isolated, and *RbcS*-*1A* is classified into group A while the rest belong to group B according to the chromosomal location [28]. Twenty *TaRbcS* family genes were identified in *Triticum aestivum* L. [29]. Five *RbcSs* family genes were identified in *Oryza sativa* L. [30]. The research on the Rubisco gene in *Brassica napus* L. (*B*. *napus* AACC, 2n = 38) remains relatively limited. Therefore, this study conducts a comprehensive bioinformatics analysis of the small subunit *RbcS* gene of Rubisco in *B*. *napus*, providing a valuable reference for further investigations into the structure and function of the *BnRbcS* gene. This analysis not only reveals the functional diversity and regulatory mechanisms of the Rubisco small subunit gene family (*RbcS*) in *B*. *napus* but also provides a theoretical foundation for identifying potential targets to optimize Rubisco activity through genetic engineering. Future research could integrate gene editing techniques with functional validation to elucidate the specific roles of *RbcS* genes in *B*. *napus* photosynthesis. Such work has the potential to offer innovative strategies for the potential improvement of photosynthetic efficiency in crop varieties.

## 2. Results

### 2.1. Genome-Wide Identification of the RbcS Family Gene Members of Brassica Species

To identify putative *RbcS* genes in six *Brassica* species, four RbcS protein sequences from *A*. *thaliana* (*Arabidopsis thaliana 10*) were used as query sequences for a BLASTP analysis via the BRAD website. Ultimately, 8, 8, 8, 8, 16, and 13 RbcS gene family members were identified in *B*. *rapa*, *B*. *nigra*, *B*. *oleracea*, *B*. *juncea*, *B*. *napus,* and *B*. *carinata* (Table 1). HMM profiles family-specific RuBisCO_ssu_N (PF12338) and RuBisCO_sc_dom (PF00101), which were extracted from the Pfam database (http://pfam.xfam.org/search, accessed on 7 August 2025), were searched against the local database using an E-value < 1 × 10^−5^. Finally, the members selected from Arabidopsis proteins and conserved domains were clustered, and the number of *RbcS* gene family members identified in each species was visualized using Venn diagrams (Figure 1). In *B*. *rapa*, the intersection of gene members represented in the Venn diagram comprised 21 genes. Further sequence alignment and confirmation of chromosomal locations revealed that a gene cluster was formed by eight of these genes. Consequently, the number of *RbcS* genes in *B*. *rapa* was confirmed to be eight.

In *B*. *napus*, the *RbcS* gene family consists of 16 members (Table 2). The expansion of gene copies may be linked to the genomic complexity of *B*. *napus* as a polyploid plant, as well as its heightened demand for photosynthetic efficiency. Among the 16 members of this gene family, eight were located on the A subgenome and eight on the C subgenome, with gene lengths varying from 802 to 3705 bp. The protein sequence alignment of *RbcS* gene family members from *A*. *thaliana*, *B*. *rapa*, *B*. *oleracea*, and *B*. *napus* revealed the loss of *RbcS*-*2B* and *RbcS*-*3B* during evolution, as they were not identified in *B*. *rapa*, *B*. *oleracea*, or *B*. *napus*. This finding further suggests that *RbcS*-*1A* and *RbcS*-*1B* are crucial for the photosynthetic processes in *B*. *napus*.

### 2.2. Phylogenetic Analysis and Classification of RbcS Genes

A phylogenetic analysis revealed that the coding sequences of the RbcS gene family were significantly conserved across species, while notable variation existed among different isoforms in gene structure and regulation. Based on phylogenetic reconstruction and protein sequence alignment, the members of the *RbcS* gene family across seven species could be categorized into four main subgroups (Figure 2). Subgroup A was particularly conserved, primarily comprising the majority of gene members from *B*. *juncea*, *B*. *carinata*, *B*. *nigra*, and *A*. *thaliana*. Subgroups B and C predominantly consisted of *RbcS* gene family members from *B*. *napus* and its ancestral species, *B*. *rapa* and *B*. *oleracea*. A phylogenetic analysis indicated that during the tetraploidization of *B*. *napus*, the *RbcS* gene family members were primarily retained through its diploid progenitors. This retention may have been facilitated by neo- or subfunctionalization, contributing to functional diversification despite an overall conservative evolutionary trajectory. Subgroup D included a few individual genes from *B*. *napus*, *B*. *juncea*, *B*. *oleracea*, and *B*. *rapa*, which exhibited a certain degree of evolutionary homology. For example, *BjuA026368* and *BolC4t26547H* were highly homologous, further indicating that the expansion of the *RbcS* gene family in *B*. *juncea* and *B*. *napus* was primarily driven by both polyploidization (whole-genome duplication) and segmental duplication events.

### 2.3. Gene Structure and Motif Compositions

To analyze the conserved features of the *RbcS* gene family, the motif, domain, and gene structures of 16 members were analyzed (Figure 3). The results showed a high degree of conservation in gene structure. Among the identified motifs, Motifs 1, 3, 5, and 8 were present in most proteins, indicating higher conservation. In contrast, Motifs 2, 4, and 6 were unique to specific members; such unique motifs may constitute distinctive features that provide clues to functional divergence, although experimental validation is required to confirm their roles. Furthermore, a conserved domain analysis confirmed that all BnRbcS proteins contained the hallmark domain of the RbcS family. In most studied cultivars, members of the *BnRbcS* family typically comprised three exons and two introns, with the exception of *BnaA07G0167600ZS*, which contained four exons and three introns. This observation further underscores the conservation of the *RbcS* gene family throughout the duplication process in *B*. *napus*.

### 2.4. Chromosomal Distribution of BnRbcS Gene

Using TBtools (2020), chromosomes were localized based on the genome annotation files concerning *B*. *rapa*, *B*. *oleracea*, and *B*. *napus* (Figure 4). We also found that 16 *BnRbcS* genes were localized on six chromosomes in *B*. *napus*. Eight *BrRbcS* genes were localized on three chromosomes (A2, A4, and A7) in the genome of *B*. *rapa*, and eight *BoRbcS* genes were localized on three chromosomes (C2, C4, and C6) in the genome of *B*. *oleracea*. Comparing the distribution patterns of *RbcS* genes in the A and C subgenomes of *B*. *napus* and in the genomes of *B*. *rapa* and *B*. *oleracea*, and given that gene counts may vary with genome version and annotation method, the results suggest that the relative positions of *RbcS* genes in *B*. *napus* have been largely conserved compared to its diploid ancestors. The results suggest that the relative positions of RbcS genes in *B*. *napus* have been largely conserved compared to its diploid ancestors.

### 2.5. Collinearity Analysis of the RbcS Gene Family

Gene duplication, mediated primarily through segmental (or whole-genome) and tandem duplication events, is a key driver of gene family expansion and genome evolution in plants [31]. To infer the history of segmental duplication events among the RbcS genes, we performed a collinearity analysis, which identified the conserved gene order across genomic regions, thereby highlighting regions derived from shared ancestral sequences. This analysis encompassed *B*. *juncea* (AABB), *B*. *napus* (AACC), *B*. *carinata* (BBCC), and their ancestral species (Figure 5). The comparison revealed extensive collinear relationships: in AABB, we identified 2 intra-genomic collinear gene pairs, 7 pairs with the AA progenitor, and 5 pairs with the BB progenitor. In AACC, we found 13 intra-genomic pairs, 15 pairs with AA, and 14 pairs with CC. For BBCC, there were 21 intra-genomic pairs, 15 pairs with BB, and 16 pairs with CC. These widespread collinear blocks strongly support the notion that segmental duplications, primarily stemming from polyploidization events, have been a major mechanism for the expansion and retention of the *RbcS* family in these Brassica allotetraploids.

### 2.6. Physicochemical Properties and Subcellular Localization of BnRbcS Gene

A physicochemical analysis of BnRbcS family protein members showed that the number of amino acids encoded by the 16 *BnRbcS* genes varied from 111 (BnaA02G0163800ZS) to 181. The molecular weight was between 12.31 and 20.37 kDa; the gene with the highest molecular weight was BnaC02T0209300ZS, and the gene with the lowest was BnaA02T0163800ZS. Members of the BnRbcS protein family had isoelectric points (pIs) ranging from 6.72 to 9.58. The instability coefficient of BnRbcS protein members ranged between 25.35 and 47.73. An instability coefficient greater than 40 was used as the threshold; thus, there was 1 stable protein in B.napus, while the remaining 15 proteins were unstable. The fatty acid coefficients of the 16 BnRbcS proteins ranged from 70.55 to 81.71, and the hydrophilicity (GRAVY) values (J) of the proteins were all less than 0, indicating that they were all hydrophilic. The subcellular localization results of 16 *BnRbcS* genes showed that 16 genes were all expressed in the chloroplast (Table 3).

### 2.7. Analysis of the Secondary Structure of RCBS Gene Family Member Proteins in B. napus

The secondary structure of proteins refers to local folding, which is the regular, repetitive spatial conformation formed in local regions through interactions such as hydrogen bonds between adjacent amino acid residues in the primary structure. The main forms include α-helix, extended chain, and random coil [32]. As illustrated in Table 4, the number of α-helices in RbcS proteins of B. napus varied between 31 and 51, constituting 23.76% to 29.67% of the total structure. In contrast, the number of extended chains ranged from 12 to 34, representing 10.81% to 18.78%. Additionally, the number of random coils varied from 69 to 113, accounting for 54.14% to 62.43% of the protein structure (Table 4).

### 2.8. Potential Interaction Prediction Analysis of BnRbcS Gene

Through the STRING online software, it was predicted and analyzed that the BnRbcS protein exhibited high homology with the RbcS1A and RbcS1B proteins in *A*. *thaliana*. By investigating the RbcS1A and RbcS1B proteins in A. thaliana, the interaction mechanism of the BnRbcS protein was elucidated (Figure 6). The results indicate that the proteins interacting with the PsRbc protein can be categorized into two groups. (1) Phosphoglycerate kinase family kinases: in the Calvin cycle, phosphoglycerate kinase functions as an energy exchange point, converting a portion of the energy from the unstable high-energy molecule 1,3-BPG (1,3-bisphosphoglycerate) into the universal ATP currency, while simultaneously generating 3-PGA, an important intermediate cargo, which prepares for the subsequent step of synthesizing the sugar building block (G3P). Without this energy conversion and substrate generation by PGK, the carbon fixation cycle cannot operate effectively. (2) The ribulose bisphosphate carboxylase, known as the Rubisco holoenzyme, consists of eight large subunits (RbcL) and eight small subunits (RbcS). The large subunit (RbcL) is primarily responsible for catalytic activity, containing both the substrate binding site and the catalytic center. Class 2 proteins competitively bind with RbcS during the processes of carbon dioxide fixation and the oxidative cleavage of pentose substrates.

### 2.9. Analysis of the Cis-Acting Elements in the Promoter Regions of the BnRbcS Genes

Cis-regulatory sequences can regulate in transcriptional regulation plant growth, thereby affecting gene expression levels [33]. We analyzed cis-acting elements in the promoter regions of *RbcS* genes in *B*. *napus*, finding elements associated with development, hormone responses, stress responses, and light responses (Figure 7). We focused on the components related to light response, which were primarily enriched in six major components, encompassing 18 motifs, suggesting a key regulatory function in light-mediated processes. The I-box motif was present in the promoters of 16 *BnRbcS* genes and constituted part of a light-responsive element. The G-box, GATA-motif, and TCT-motif were found in the majority of promoters, with the exceptions being *BnaA04G0100700ZS*, *BnaC06G0156800ZS*, *BnaA07G0167600ZS*, *BnaC04G0420700ZS*, and *BnaC02G0209800ZS*, all of which were associated with light-responsive elements. This observation further suggests that members of the *RbcS* gene family play a significant role in the process of photosynthesis. This analysis result is consistent with existing studies, which indicate that the *RbcS* promoter contains light-responsive elements and is light-induced at the transcriptional level [34].

### 2.10. Analysis of BnRbcS Gene Expression Patterns

To further reveal the expression patterns and potential biological functions of the identified *BnRbcS* genes, we analyzed the expression of *BnRbcS* genes across different tissues based on the RNA-seq data obtained throughout the entire reproductive development period of *B*. *napus* (Figure 8). The results showed that most of the *BnRbcS* genes were highly expressed in the fertility stage. A heatmap analysis revealed distinct expression patterns for each gene in the family. The *BnaA04G0101200ZS* and *BnaC04G0380700ZS* exhibited high expression patterns in different tissues, where the expression levels were significantly higher than other genes, especially in the leaves and siliques. These results suggest that only a few members might be essential, which could imply potential functional redundancywithin the gene family.

### 2.11. Analysis of BnRbcS Gene Expression Patterns Under Light Treatment at 6 h and 12 h

The *BnRbcS* genes play crucial roles in the photosynthetic. To investigate their light response, we subjected 16 family members to light treatment and assessed their expression in leaves after 6 h and 12 h using a qRT-PCR. As shown in Figure 9, among the 16 *BnRBcS* genes, 10 genes showed significant differences in expression levels under two different light treatments. Among them, the expression levels of *RbcSA2-3*, *RbcSA2-3*, and *RbcSC2-1* were significantly higher than those of other genes, except for *RbcSA2-2*, and nine genes exhibited significantly higher expression at 12 h compared to 6 h, suggesting that their expression was enhanced under prolonged light exposure within this timeframe under 6–12 h. In contrast, five genes displayed no significant difference in expression levels between the two treatments. A further expression analysis of the 16 genes under 6 h treatment revealed no significant differences among them, whereas significant differences were observed among multiple *BnRbcS* genes under 12 h treatment. Based on the analysis of the cis-acting elements in the BnRbcS promoter, multiple photoresponsive elements were identified in the promoter of this gene family member. Therefore, we hypothesize that the promoter type of this gene family may belong to light-inducible promoters.

### 2.12. Haplotype Analysis of RbcSA4-1

Based on gene expression profiling, two *RbcS* gene family members, *RbcSA4-1* and *RbcSC4-1*, were identified as highly expressed in the leaves and siliques of *B*. *napus* (Table 5, Figure 10). Rubisco enzyme activity was measured in leaves and siliques across a natural population of 191 accessions, and a haplotype analysis was performed for these two highly expressed genes. For *RbcSA4-1*, four haplotypes (Hap1–Hap4) were identified. Accessions carrying Hap1, Hap2, or Hap3 exhibited significantly higher Rubisco activity compared to those with Hap4, suggesting that these haplotypes represent favorable variants for the preliminary screening of high photosynthetic efficiency germplasm. These haplotypes thus represent promising candidates for the preliminary screening of germplasm with potentially high photosynthetic efficiency; however, their functional impact requires direct validation in future studies.

## 3. Discussion

Rubisco, a dual-functional enzyme predominantly localized in the chloroplast stroma of plants, exhibits two distinctive characteristics: substrate promiscuity and low catalytic efficiency. It catalyzes both carboxylation and oxygenation reactions; however, it displays a poor affinity for atmospheric CO_2_ and a slow turnover rate. To compensate for its inefficiency, organisms synthesize large amounts of this enzyme. In photosynthesis, Rubisco mediates carbon oxidation during photorespiration and contributes to carbon loss during CO_2_ fixation, thereby critically influencing the net photosynthetic rate of plants [35]. Previous studies have shown that in higher plants, the *RbcS* gene family is comprised of between 2 and 22 members, with the number varying among species [36,37]. For instance, a soybean contains at least ten members [38], wheat contains twenty members, and rice contains five members [39]. In this study, 61 *RBCS* genes were identified in the U triangle consisting of three fundamental diploid species: *B*. *rapa* (AA, 2n = 20), *B*. *nigra* (BB, 2n = 16), and *B*. *oleracea* (CC, 2n = 18), and three amphidiploids: *B*. *napus* (AACC, 2n = 38), *B. juncea* (AABB, 2n = 36), and *B. carinata* (BBCC, 2n = 34). These included 8, 8, and 8 genes in the diploid progenitors *B. rapa*, *B. nigra*, and *B. oleracea* and 8, 16, and 13 *RbcS* genes in the allotetraploid *B. juncea*, *B*. *napus*, and *B. carinata*. We found that in allopolyploid species, the number of RbcS genes does not increase exponentially with genome duplication. We speculate that this may be due to the following reasons. (1) Gene loss/silencing: in the early stage of polyploid formation, there are a large number of functionally redundant gene copies. The most efficient way to streamline the genome is to randomly or selectively lose one copy. (2) In allotetraploid, the two sets of genomes from two ancestors often have unequal status, which is usually due to the bias of epigenetic regulation. We identified 16 *RbcS* genes in *B. napus*, which are homologous to *A*. *thaliana RbcS*-*1A* and *RbcS*-*1B*. Notably, *RbcS*-*2B* and *RbcS*-*3B* were absent in both *B. napus* and its progenitor species.

A phylogenetic analysis classified the 16 *RbcS* genes in *B. napus* into five distinct clades. Combined with an analysis of protein physicochemical properties, 13 members were found to contain 181 amino acids, while the remaining three genes BnaC02T0209800ZS, BnaA07T0167600ZS, and BnaA02T0163800ZS encode proteins of 174, 128, and 111 amino acids, respectively. These results suggest that the nucleotide sequences of most RbcS family members are conserved, and the three atypical genes may have undergone mutation during polyploidization, potentially reflecting functional diversification. In a related context, Ruuska et al. [40] reported that Rubisco in *B. napus* participates in a metabolic pathway that channels carbon into oil, maximizing the efficiency of carbon storage as oil.

Notably, in this study, the *RbcS* gene family members in *B. napus* exhibited a pan-tissue expression pattern, though their relative expression levels varied considerably among different members. Among them, *RbcSA4-1* and *RbcSC4-1* showed notably high expression in leaves and siliques (Appendix A), suggesting that photosynthetic product accumulation and biomass formation in rapeseed largely depend on photosynthetic activities in these two organs.

Furthermore, when *BnRbcS* family members were subjected to 6 h and 12 h light treatments, expression levels among members showed no significant differences after 6 h, whereas pronounced divergence in expression profiles was observed after 12 h. Combined with the promoter sequence analysis, we hypothesize that the presence of light-responsive cis-elements may be a key factor contributing to the differential expression under prolonged illumination.

In addition, a haplotype analysis of *RbcSA4-1* was conducted in a natural population consisting of 191 accessions. Four haplotypes (Hap1–Hap4) were identified, among which accessions carrying Hap1, Hap2, and Hap3 exhibited significantly higher Rubisco enzyme activity compared to those with Hap4. This indicates that Hap1, Hap2, and Hap3 could serve as superior haplotypes for the preliminary screening of high photosynthetic efficiency materials.

## 4. Materials and Methods

### 4.1. Genome-Wide Identification of the RbcS Gene Family

We sought to identify members of the *RbcS* gene family in *Brassica napus* L. (*B. napus*, AACC, 2n = 36), and related Brassica species: *Brassica rapa* L. (*B. rapa*, AA, 2n = 20), *Brassica nigra* L. (*B. nigra*, BB, 2n = 16), *Brassica oleracea* L. (*B. oleracea*, CC, 2n = 18), *Brassica juncea* L. (*B. juncea*, AABB, 2n = 36), and *Brassica carinata* L. (*B. carinata*, BBCC, 2n = 34). We initially employed the four RBCS protein sequences from A. thaliana as query sequences for the BLASTP analysis via the BRAD website (E-value ≤ 1 × 10^−5^) [29,41,42]. Subsequently, we accessed the genome database and protein reference sequences, along with their respective annotation files for the six species from the BRAD database, creating a local database populated with the RBCS proteins. In the second step, we obtained the HMM profiles for the family-specific RuBisCO_ssu_N (PF12338) and RuBisCO_sc_dom (PF00101), which represent the conserved domains of the RbcS gene family, extracted from the Pfam database (http://pfam.xfam.org/search, accessed on 5 August 2025). Finally, we utilized a Venn diagram to illustrate the union of the identified RBCS proteins as members of the RBCS gene family for further analysis.

### 4.2. Multiple Sequence Alignment and Phylogenetic Analysis of the RbcS Gene Family

Multiple sequence alignment of RbcS protein sequences from seven species, *B. rapa*, *B. nigra*, *B. oleracea*, *B. juncea*, *B. napus*, *B. carinata*, and *A*. *thaliana*, was performed using MEGA 11.0 [43]. A phylogenetic tree was then constructed with the maximum likelihood (ML) method, employing 1000 bootstrap replicates to assess node support. The resulting tree was visualized and annotated using the EvolView platform (https://evolgenius.info/evolview, accessed on 5 August 2025).

### 4.3. Gene Structure and Conserved Motif Analysis of BnRbcS

Gene annotation data for BnRbcS family members were obtained from the BnTIR database (http://yanglab.hzau.edu.cn/, accessed on 5 August 2025). Gene structures were visualized using the online Gene Structure Display Server (GSDS; http://gsds.gao-lab.org/, accessed on 13 August 2025) [44]. Conserved protein motifs were identified with the MEME online tool (http://meme-suite.org/tools/meme, accessed on 13 August 2025) [45], with the motif count set to 10 and other parameters kept as the default. All results were integrated and visualized using TBtools [46].

### 4.4. Chromosomal Distribution and Collinearity Analysis of BnRbcS

*B. napus*, an allotetraploid species, has undergone extensive genomic rearrangements during its evolution. Gene duplication events, including segmental and tandem duplications, were investigated as key drivers of gene family expansion [47]. Genomic sequences and annotation files (GFF3 format), the *B. napus*, *B. rapa*, and *B. oleracea*, were downloaded from the BRAD database website (www.brassicadb.cn/#/Download/, accessed on 5 August 2025) and chromosomal locations mapped using TBtools. TBtools was employed to investigate the gene duplication mechanisms and collinearity within the *RbcS* gene family in *B. napus*, *B. carinata*, *B. juncea*, and its diploid progenitors [48].

### 4.5. Physicochemical Properties and Subcellular Localization of BnRbc Proteins

Amino acid sequences of all BnRbcS proteins were analyzed using the ExPASy ProtParam tool (http://web.expasy.org/protparam/, accessed on 13 August 2025) [49] to predict physicochemical properties, including amino acid composition, molecular weight, theoretical pI, instability index, aliphatic index, and GRAVY value, and subcellular localization was also predicted.

### 4.6. Cis-Acting Element Analysis in BnRbcS Gene Promoters

The 2000 bp genomic sequences upstream of the transcription start sites of *BnRbcS* genes were extracted by TBtools. Putative cis-acting regulatory elements were identified using the PlantCARE database (http://bioinformatics.psb.ugent.be/webtools/plantcare/html/, accessed on 12 August 2025) [50]. Results were compiled and visualized with TBtools.

### 4.7. Protein–Protein Interaction Prediction

The proteins sequences of RbcS-1A and RbcS-1A from *A*. *thaliana* were submitted to the STRING database, for the prediction and visualization of protein–protein interaction networks [51,52].

### 4.8. Tissue-Specific Expression Analysis of BnRbcS

The expression profiles of *BnRbcS* genes across different tissues were analyzed using RNA-seq data from the BnIR database (http://yanglab.hzau.edu.cn/BnIR, accessed on 16 August 2025) and visualized as a heatmap with TBtools. Based on the expression patterns, two highly expressed members in leaves *BnaA04G0101200ZS* and *BnaC04G0380700ZS* were selected for further study. Rubisco enzyme activity was measured in leaves and siliques of a natural population comprising 191 accessions, and a haplotype analysis was performed for the candidate genes.

### 4.9. qRT-PCR Analysis Under 6-h and 12-h Light Treatments

To investigate the response of *BnRbcS* gene family members to photoperiod duration, we subjected ZS11 seedlings at the five-leaf seedling stage to 6 h and 12 h light treatments in a growth chamber (the model of the growth chamber is SPX-400-GB (Shanghai Langdi Testing Equipment Co., Ltd., Shanghai, China); temperature: 25 ± 1.5 °C; relative humidity: 50%; light intensity: 20000LX), and quantified their expression levels in leaves using a qRT-PCR analysis. Specific primers were designed using the Primer Quest Tool (http://sg.idtdna.com/Primerquest/Home/Index, accessed on 5 August 2025), and priority was given to selecting primers close to the 3′ end of the gene. *ACTIN* was used as an internal reference gene in this experiment. The gene abbreviation ID and primer sequences are shown in Appendix A. The qRT PCR quantitative experiment was conducted using the reagent kit from Shanghai Yeasen Biotechnology Co., Ltd. (Shanghai, China), first strand reverse transcription was completed using Hifair ^®^ III 1st Strand cDNA Synthesis SuperMix for qPCR (gDNA digester plus), and the first chain synthesis kit (11141es60) used Hieff for fluorescence quantitative detection with the ^®^ QPCR SYBR Green Master Mix (Low Rox Plus) kit. (All kit from Shanghai Yeasen Biotechnology Co., Ltd., Shanghai, China).

## 5. Conclusions

In this study, 61 *RbcS* genes were identified in members of the U triangle: *B. napus*, and related Brassica species: *Brassica rapa* L., *Brassica nigra* L., *Brassica oleracea* L., *Brassica juncea* L., and *Brassica carinata* L. A phylogenetic tree analysis further elucidated the homologous relationships between the *BnRbcS* gene family and *RbcS* gene family members in other species, offering significant insights into their functional evolution and species specificity. Following their identification in the *B. napus*, the 16 RbcS genes were subjected to a comprehensive analysis of their physicochemical properties, phylogeny, chromosomal distribution, gene structure, cis-acting elements, and expansion patterns. In the expression analysis, the results showed that most of *BnRbcS* genes were highly expressed in the fertility stage and presented a pan organizational expression pattern. We exposed 16 *BnRbcS* family genes to light treatment and assessed their expression levels in rapeseed leaves after 6 and 12 h of light exposure using a real-time fluorescence quantitative PCR. In total, 10 genes exhibited significantly higher expression levels under 12 h of light treatment compared to 6 h, suggesting that 12 h of light treatment can enhance the expression of these genes. A further expression analysis of the 16 genes under 6 h treatment revealed no significant differences among them, whereas significant differences were observed among multiple *BnRbcS* genes under 12 h treatment. In addition, a haplotype analysis of *BnaA04G0101200ZS* was conducted in a natural population consisting of 191 accessions. This indicated that Hap1, Hap2, and Hap3 could serve as superior haplotypes for the preliminary screening of high photosynthetic efficiency materials.

## Figures and Tables

**Figure 1 plants-15-00058-f001:**
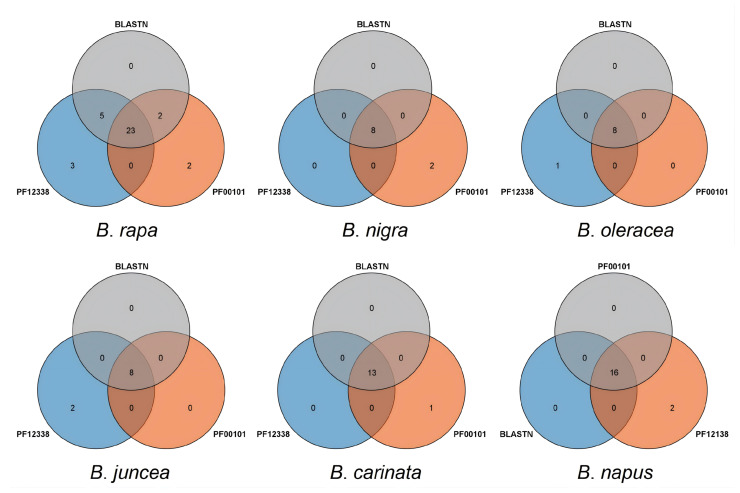
Venn diagram for the identification of gene family members. Notes: Gray, blue, and orange represent members identified in six Brassica species using the RBCS protein sequence from *A*. *thaliana*, the Pfam domain PF00101, and PF12338, which were subsequently integrated in a Venn diagram.

**Figure 2 plants-15-00058-f002:**
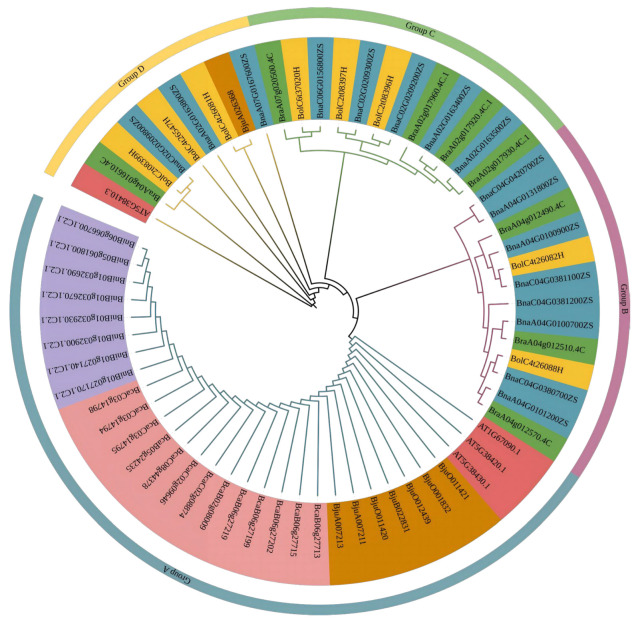
Phylogeny of the *RbcS* family members in *B. napus*, *B. rapa*, *B. oleracea*, and *A*. *thaliana*. Notes: The colors on the outer ring of the phylogenetic tree indicate the assigned subgroups: cyan (#76a5af) represents Group A, purple (#c27ba0) represents Group B, light green (#93c47d) represents Group C, and yellow (#ffd966) represents Group D. The colors of the branch labels denote RbcS family members from different species: red (#e06666), green (#6aa84f), yellow (#fec232), blue (#5da0af), tan (#ce7e00), pink (#ea9999), and lavender (#64a7db) correspond to *A*. *thaliana*, *B. rapa*, *B. oleracea*, *B. napus*, *B. juncea*, *B. carinata*, and *B. nigra*.

**Figure 3 plants-15-00058-f003:**
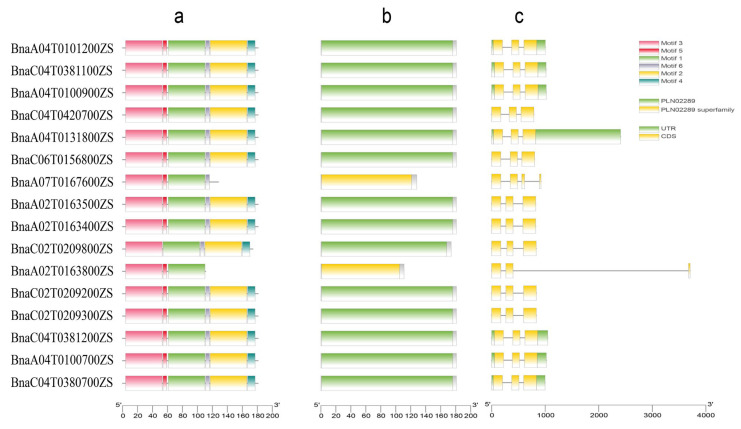
The visual display of the *BnRbcS* genes motifs (**a**), domains (**b**), and structure (**c**).

**Figure 4 plants-15-00058-f004:**
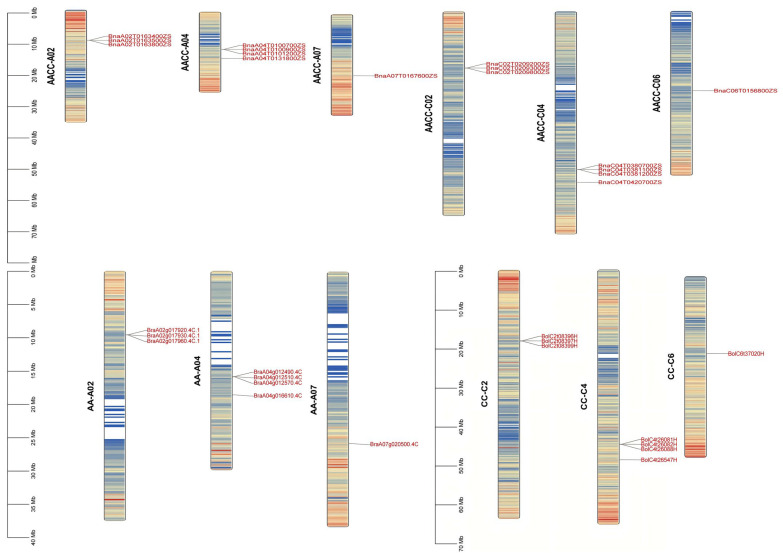
Distribution of *RbcS* family numbers on the chromosomes of *B. napus*, *B. rapa*, and *B. oleracea*. Notes: AACC, AA, CC stand for *B. napus*, *B. rapa*, and *B*. *oleracea.* Notes: From blue to red, it indicates that the density of genes on the chromosome increases from small to large. The darker the red, the higher the density of genes in this segment of the chromosome.

**Figure 5 plants-15-00058-f005:**
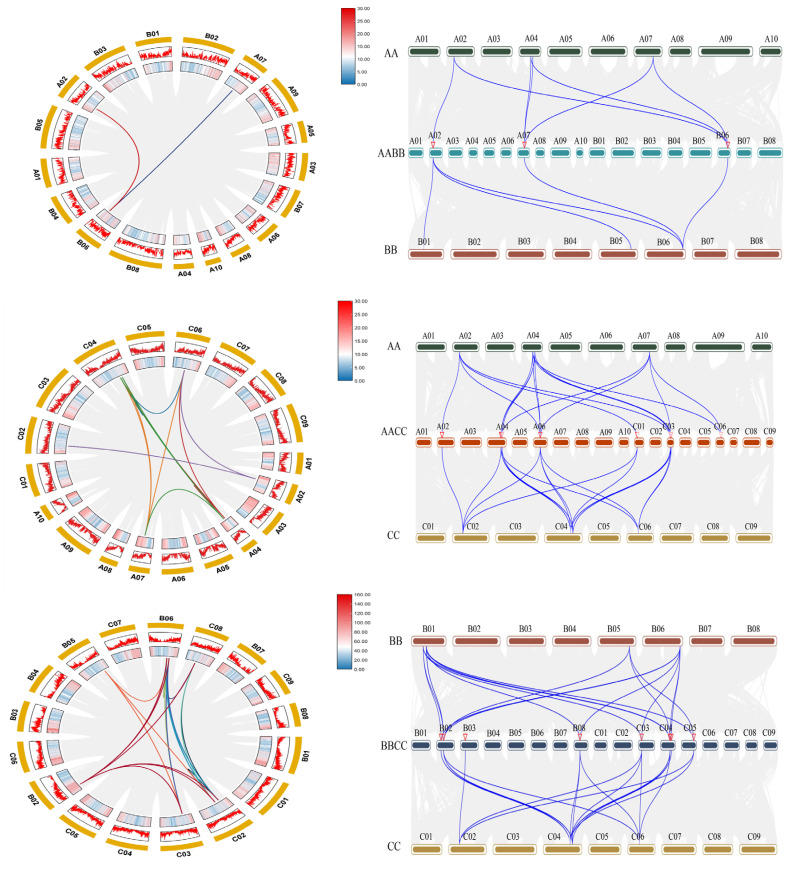
Collinearity analysis of the *RbcS* gene family in *B. juncea*, *B. napus*, *B. carinata*, and its diploid ancestors. Notes: The three circular graphs on the left represent intra-specific collinearity within the *B. juncea* (AABB), *B. napus* (AACC), and *B. carinata* (BBCC) genomes, with gray lines representing all collinear gene pairs within the genome and colored lines representing collinear gene pairs among Rbcs gene members; The three figures on the right represent the inter species collinearity between the *B. juncea*, *B. napus*, and *B. carinata* genomes and their ancestral species. The gray lines represent all collinear gene pairs between genomes, while the colored lines represent collinear gene pairs between Rbcs gene members.

**Figure 6 plants-15-00058-f006:**
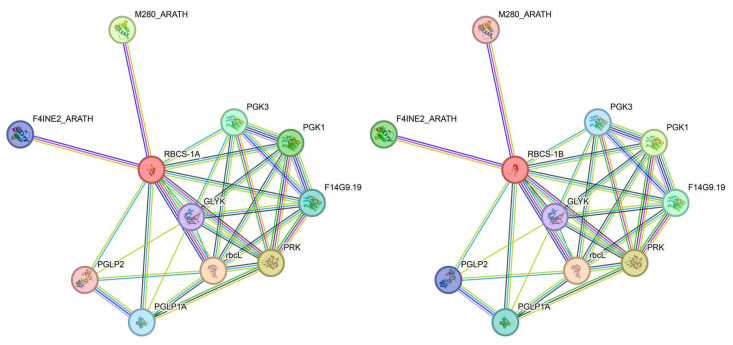
Prediction network of proteins interacting with BnRbcS.

**Figure 7 plants-15-00058-f007:**
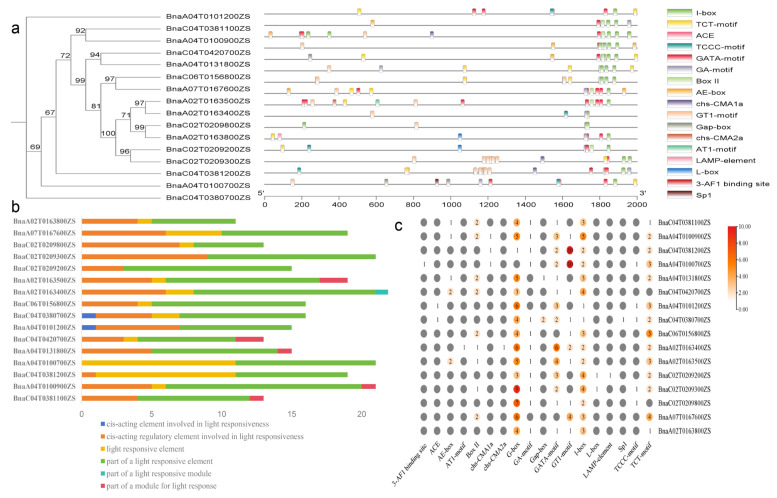
Analysis of cis-acting elements in the promoter regions of *BnRbcS* genes. Notes: (**a**) The distribution of photosynthetic cis-acting elements on the promoter of *BnRbcS* members; (**b**) the response pathway of the cis-acting elements of the *BnRbcS* member promoter; (**c**) visualization of the number of cis elements in the promoter of *BnRbcS* members.

**Figure 8 plants-15-00058-f008:**
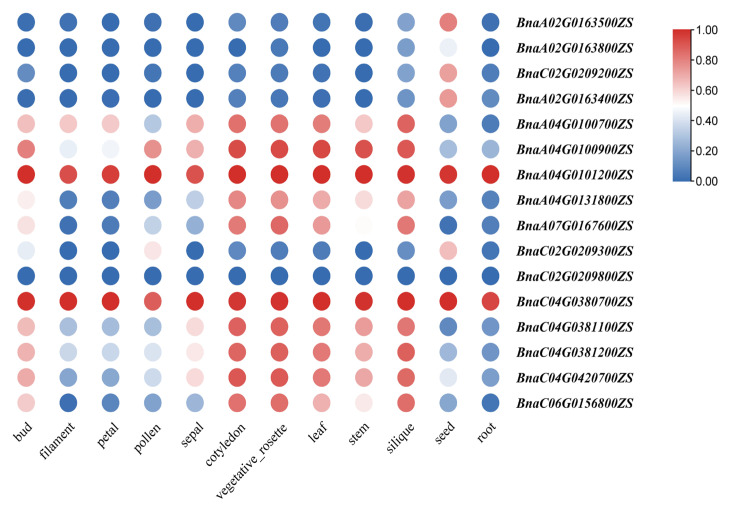
Heatmap analysis of BnRbcS gene family members’ expression.

**Figure 9 plants-15-00058-f009:**
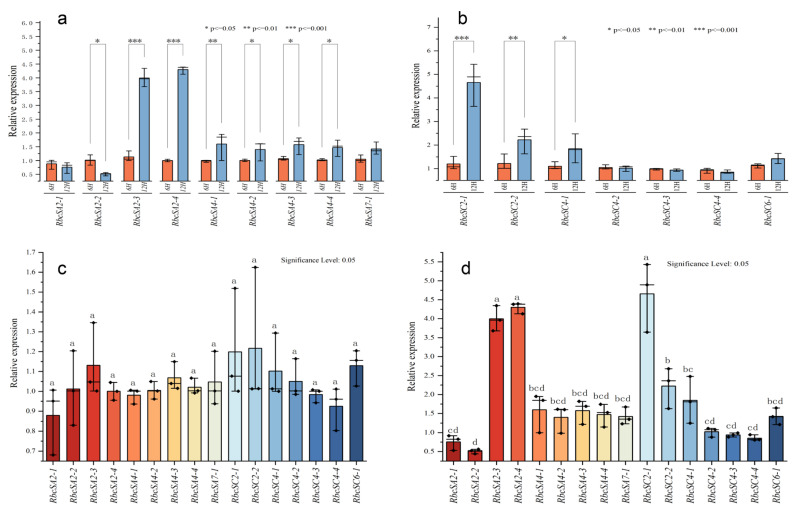
Expression analysis of *BnRbcS* genes under light treatment at 6 h and 12 h. Notes: Panels (**a**,**b**) show the expression levels of *BnRbcS* genes located on the A subgenomes and C subgenomes, respectively. Panels (**c**,**d**) display the expression changes of the 16 BnRbcS gene members under 6 h and 12 h light treatments, respectively. Different lowercase letters indicate significant differences between treatments at *p* ≤ 0.05.

**Figure 10 plants-15-00058-f010:**
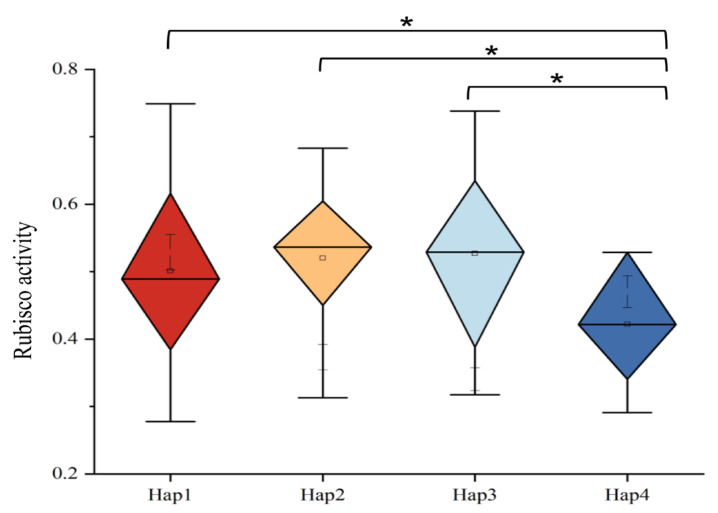
Haplotype analysis of *RbcSA4-1. Notes:* *: significantdifferences between treatments at *p* ≤ 0.05.

**Table 1 plants-15-00058-t001:** Number of *RbcS* gene family members in tetraploid rapeseed and its ancestral species.

Genome ID	Plant Species	Genome Type	Genome Size (Kb)	Coding Genes	RcbS Genes
Chiifu.v4	*Brassica rapa* ssp. *pekinensis*	AA	113,126	83,682	8
CN115125v1	*Brassica nigra*	BB	532,791	67,021	8
HDEM.v0	*Brassica oleracea* var. *italica*	CC	551,006	61,279	8
tumida.v1.5	*Brassica juncea* var. *tumida*	AABB	915,101	79,644	8
ZS11.v0	*Brassica napus* ssp. *oleifera*	AACC	987,244	100,919	16
zd-1.v0	*Brassica carinata*	BBCC	1,082,762	97,149	13

**Table 2 plants-15-00058-t002:** Identification of *RbcS* family gene members from *A*. *thaliana*, *B. rapa*, *B. oleracea*, and *B. napus*.

*A. Thaliana* Homologues	*B. rapa* Homologues	*B. oleracea* Homologues	*B. napus* Homologues	Chr	Start	End	Length
*AT1G67090/RbcS-1A*	*BraA02g017930.4C*		*BnaA02G0163500ZS*	A02	9,675,456	9,676,279	823
*AT1G67090/RbcS-1A*	*BraA02g017960.4C*		*BnaA02G0163800ZS*	A02	9,686,731	9,690,436	3705
*AT1G67090/RbcS-1A*		*BolC2t08396H*	*BnaC02G0209200ZS*	C02	17,993,607	17,994,441	834
*AT5G38430/RbcS-1B*	*BraA02g017920.4C*		*BnaA02G0163400ZS*	A02	9,672,337	9,673,159	822
*AT5G38430/RbcS-1B*	*BraA04g012510.4C*		*BnaA04G0100700ZS*	A04	11,915,596	11,916,617	1021
*AT5G38430/RbcS-1B*	*BraA04g012490.4C*		*BnaA04G0100900ZS*	A04	11,923,776	11,924,795	1019
*AT5G38430/RbcS-1B*	*BraA04g012570.4C*		*BnaA04G0101200ZS*	A04	11,938,835	11,939,838	1003
*AT5G38430/RbcS-1B*	*BraA04g016610.4C*		*BnaA04G0131800ZS*	A04	14,773,357	14,775,764	2407
*AT5G38430/RbcS-1B*	*BraA07g020500.4C*		*BnaA07G0167600ZS*	A07	19,554,954	19,555,875	921
*AT5G38430/RbcS-1B*		*BolC2t08397H*	*BnaC02G0209300ZS*	C02	17,997,780	17,998,614	834
*AT5G38430/RbcS-1B*		*BolC2t08399H*	*BnaC02G0209800ZS*	C02	18,074,145	18,074,977	832
*AT5G38430/RbcS-1B*		*BolC4t26088H*	*BnaC04G0380700ZS*	C04	50,500,677	50,501,674	997
*AT5G38430/RbcS-1B*		*BolC4t26082H*	*BnaC04G0381200ZS*	C04	50,539,319	50,540,335	1016
*AT5G38430/RbcS-1B*		*BolC4t26081H*	*BnaC04G0381200ZS*	C04	50,546,522	50,547,569	1047
*AT5G38430/RbcS-1B*		*BolC4t26547H*	*BnaC04G0420700ZS*	C04	54,553,194	54,553,982	788
*AT5G38430/RbcS-1B*		*BolC6t37020H*	*BnaC06G0156800ZS*	C06	25,387,399	25,388,201	802
*AT5G38420/RbcS-2B*			/	/	/	/	/
*AT5G38410/RbcS-3B*			/	/	/	/	/

**Table 3 plants-15-00058-t003:** Physicochemical properties and subcellular localization of *BnRbcS* genes.

Sequence ID	Number of Amino Acid	Molecular Weight	Theoretical pI	Instability Index	Aliphatic Index	Grand Average of Hydropathicity	Subcellular Localization
BnaC04T0381100ZS	181	20,183.17	8.23	32.21	73.26	−0.16	chloroplast
BnaA04T0100900ZS	181	20,183.17	8.23	32.21	73.26	−0.16	chloroplast
BnaC04T0381200ZS	181	20,199.17	8.23	33.27	72.71	−0.175	chloroplast
BnaA04T0100700ZS	181	20,227.22	8.23	32.8	73.76	−0.161	chloroplast
BnaA04T0131800ZS	181	20,158.12	8.23	31.61	72.21	−0.16	chloroplast
BnaC04T0420700ZS	181	20,185.14	8.23	31.61	72.21	−0.175	chloroplast
BnaA04T0101200ZS	181	20,303.32	8.23	31.74	73.76	−0.164	chloroplast
BnaC04T0380700ZS	181	20,317.3	7.59	30.78	74.31	−0.16	chloroplast
BnaC06T0156800ZS	181	20,302.31	8.23	29.04	70.55	−0.223	chloroplast
BnaA02T0163400ZS	181	20,324.46	8.48	33.66	72.76	−0.162	chloroplast
BnaA02T0163500ZS	181	20,323.48	8.69	32.83	72.76	−0.162	chloroplast
BnaC02T0209200ZS	181	20,462.64	8.48	29.53	72.21	−0.186	chloroplast
BnaC02T0209300ZS	181	20,377.46	7.58	37.81	75.47	−0.157	chloroplast
BnaC02T0209800ZS	174	19,560.59	8.7	36.41	75.69	−0.127	chloroplast
BnaA07T0167600ZS	128	13,981.96	6.72	25.35	77.66	−0.103	chloroplast
BnaA02T0163800ZS	111	12,311.44	9.58	47.73	81.71	−0.089	chloroplast

**Table 4 plants-15-00058-t004:** Analysis of the secondary protein structure of *BnRbcS* genes.

Protein ID	Alpha Helix (Hh)		Extended Strand (Ee)		Random Coil (Cc)	
	Quantity	Proportion (%)	Quantity	Proportion (%)	Quantity	Proportion (%)
BnaC04T0381100ZS	50	27.62%	30	16.57%	101	55.80%
BnaA04T0100900ZS	50	27.62%	30	16.57%	101	55.80%
BnaC04T0381200ZS	47	25.97%	33	18.23%	101	55.80%
BnaA04T0100700ZS	43	23.76%	34	18.78%	104	57.46%
BnaA04T0131800ZS	45	24.86%	23	12.71%	113	62.43%
BnaC04T0420700ZS	51	28.18%	31	17.13%	99	54.70%
BnaA04T0101200ZS	48	26.52%	33	18.23%	100	55.25%
BnaC04T0380700ZS	47	25.97%	33	18.23%	101	55.80%
BnaC06T0156800ZS	50	27.62%	29	16.02%	102	56.35%
BnaA02T0163400ZS	47	25.97%	31	17.13%	103	56.91%
BnaA02T0163500ZS	50	27.62%	22	12.15%	109	60.22%
BnaC02T0209200ZS	47	25.97%	28	15.47%	106	58.56%
BnaC02T0209300ZS	50	27.62%	33	18.23%	98	54.14%
BnaC02T0209800ZS	48	27.59%	31	17.82%	95	54.60%
BnaA07T0167600ZS	38	29.69%	18	14.06%	72	56.25%
BnaA02T0163800ZS	30	27.03%	12	10.81%	69	62.16%

**Table 5 plants-15-00058-t005:** Variation site analysis of *RbcSA4-1* haplotype.

References allele	G	A	TAT	TCTATGT	T	G
Alternative allele	A	-	-	-	-	A
SNP annotation	UTR3	UTR3	UTR3	intronic	intronic	exonic
SNP positions	11,938,853	11,938,916	11,938,973	11,939,546	11,939,632	11,939,677
Hap_1	A/A	-/-	-/-	-/-	-/-	G/G
Hap_2	G/G	A/A	TAT/TAT	TCTATGT/TCTATGT	T/T	G/G
Hap_3	G/G	A/A	TAT/TAT	TCTATGT/TCTATGT	-/-	A/A
Hap_4	G/G	A/A	-/-	TCTATGT/TCTATGT	-/-	A/A

## Data Availability

All data generated or analyzed during this study are included in this article and its Appendix A files.

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
