# Peer review of "Genome-Wide Analysis of the *RbcS* Gene Family and Expression Analysis Under Light Response in *Brassica napus* L."

_plants, 2025, doi:10.3390/plants15010058_

Round 1
Reviewer 1 Report
Comments and Suggestions for Authors
In this study, the authors focus on RuBisCO small subunit (RbcS) gene family, which is important for enhancing photosynthetic efficiency across Brassica. They analyzed phylogenetic relationships, gene structures, conserved domains, collinearity, cis-regulatory elements, expression profiles, and haplotype variations. Furthermore, they found that BnRbcS is light-responsive genes, which could be divided into long-day responsive genes, light-insensitive genes and short-day responsive gene. They also identified a superior allele for BnaA04G0101200ZS. Above results will aid in guiding high-photosynthetic-efficiency rapeseed breeding. This is an interesting study and most of the results are presented clearly. However, I still have a couple of questions and suggestions to improve this manuscript:
- Line10, 13: numbers 2 and 3 are missing.
- Line 122: napus, B. carinata ‘and’ is missing.
- Line 124: ‘using an E-value cutoff < 1.0’. I feel that the threshold value is set too high.
- Line 191: ‘between 12.3 1 and 20.37 kDa’ should be revised to ‘between 12.31 and 20.37 kDa’.
- Figure 1, 2, 4, 6, 9: Please improve image quality and clarity. The figure legends need more details.
- Figure 3, 5, 7: Please add more annotations in the figure legends.
- Figure 10: What does the ‘Rubiso’ mean in the Y axis?
- Figure 9: Please add A, B, C for each sub-figures. Check all figures.
- Table 2: What’s the unit in the columns of ‘Start’, ‘End’ and ‘Length’?
- Table 3: ‘subcellular localization’ >> ‘Subcellular Localization’
- The capitalization of the title should be consistent in all Tables.
- Figure 8: What’s the unit of gene expression?
- Table 5: Please revise the title in the header row.
- Line 468: Delete one ‘Table S1’.
- Line 487-488: This study includes more abbreviations.
- References: the format of literatures is inconsistent. Please check one by one.
Author Response
Comments1:Line10, 13: numbers 2 and 3 are missing.
Response1:We thank the reviewer for pointing out this oversight. The missing numbers "2" and "3" have been added on Lines 10 and 13, respectively, in the revised manuscript.
Comments 2:Line 122: napus, B. carinata ‘and’ is missing.
Response 2:We thank the reviewer for the careful correction. The missing conjunction "and" has been added between "B. napus" and "B. carinata" in Line 132 of the revised manuscript.
Comments 3:Line 124: ‘using an E-value cutoff < 1.0’. I feel that the threshold value is set too high.
Response 3:We sincerely apologize for this error and thank the reviewer for highlighting it. We have re-examined our gene identification process. The correct E-value < 1 × 10⁻⁵, not < 1.0. The text in Line 135 has been revised accordingly.
Comments 4:Line 191: ‘between 12.3 1 and 20.37 kDa’ should be revised to ‘between 12.31 and 20.37 kDa’.
Response 4:Thank you for catching this typographical error. The range in Line 247 has been corrected to “between 12.31 and 20.37 kDa” as suggested.
Comments 5:Figure 1, 2, 4, 6, 9: Please improve image quality and clarity. The figure legends need more details.
Response 5:We thank the reviewer for the suggestion. Figure 1 (Line145),Figure 2 (Lines 181-189), Figure 4 (Lines 218-220), Figure 6 (Line 285), and Figure 9 (Lines 337-342) have been re-drawn to improve their overall quality and clarity. Additionally, more detailed annotations have been added to their respective figure legends as requested.
Comments 6:Figure 3, 5, 7: Please add more annotations in the figure legends.
Response 6:e thank the reviewer for the suggestion. Additional annotations and explanatory details have been added to the legends of Figure3 (Line 204), Figure5 (Lines 236-243), and 7 (Lines 305-308) to improve clarity.
Comments7:Figure 10: What does the ‘Rubiso’ mean in the Y axis?
Response 7:We thank the reviewer for pointing this out. The label “Rubiso” on the Y-axis of Figure 10 is a typographical error and should “Rubisco activity”.
Comments 8:Figure 9: Please add A, B, C for each sub-figures. Check all figures.
Response 8:Thank you for the suggestion. We have re-drawn Figure 9 (Lines 337-342) and added the labels a, b c and d to each sub-figure as requested.
Comments 9:Table 2: What’s the unit in the columns of ‘Start’, ‘End’ and ‘Length’?
Response 9:The unit for ‘Start’, ‘End’, and ‘Length’ in Table 2 is base pairs (bp), as now indicated in the table.
Comments 10:Table 3: ‘subcellular localization’ >> ‘Subcellular Localization’
Response 10:We appreciate the reviewer’s attention to formatting detail. We have corrected the column title in Table 3 to “Subcellular Localization” and have ensured consistency in the capitalization of all table and figure titles throughout the manuscript.
Comments 11:The capitalization of the title should be consistent in all Tables.
Response 11:Thank you for pointing out this inconsistency. We have reviewed the titles of all tables in the manuscript and ensured consistent capitalization throughout.
Comments 12:Figure 8: What’s the unit of gene expression?
Response 12:As the expression values in Figure 8 are unitless fold changes calculated by the 2^(-ΔΔCt) method, the Y-axis has been relabeled to “Relative Expression” for clarity.
Comments 13:Table 5: Please revise the title in the header row.
Response 13:Thank you for the suggestion. The title in the header row of Table 5 has been revised to “Table 5. Variation site analysis of BnaA04G0101200ZS haplotype” as requested.
Comments 14:Line 468: Delete one ‘Table S1’.
Response 14:Thank you, the duplicate “Table S1” on Line 520 has been removed.
Comments 15:Line 487-488: This study includes more abbreviations.
Response 15:Thanks, the abbreviations have been listed as suggested.
Comments 16:References: the format of literatures is inconsistent. Please check one by one.
Response 16:We thank the reviewer for this important reminder. We have carefully checked all references in the manuscript one by one and revised them to ensure consistency and accuracy in formatting throughout the reference list.
Reviewer 2 Report
Comments and Suggestions for Authors
I have carefully reviewed the manuscript and found it to be a comprehensive and valuable contribution to understanding the RbcS gene family and its role in photosynthesis in rapeseed. The study integrates phylogenetic, structural, expression, and haplotype analyses, providing insights into the potential functional roles of BnRbcS genes. Nevertheless, several aspects could be improved to enhance scientific rigor, clarity, and readability.
Lines 21–23: The statement “enhancing photosynthetic efficiency represents a key approach for improving crop yield and biomass” is generally correct. However, it should be clarified that increased photosynthetic efficiency may lead to higher biomass, but its impact on grain or economic yield depends on carbon allocation between plant organs and is not always direct.
Lines 22–23: The description of RbcS as encoding “core components of Rubisco” is misleading. RbcS does not catalyze CO₂ fixation; the catalytic activity resides in RbcL. It would be more accurate to state that RbcS “stabilizes and regulates Rubisco assembly and activity.”
Lines 24–26: There are punctuation errors in the species names (e.g., “Brassica. napus” should be “Brassica napus L.”).
Lines 27–29: The phrase “functional significance” should be tempered; bioinformatic and expression analyses suggest “potential functional roles” rather than proven significance.
Lines 30–32: Light response experiments require details of light intensity, control conditions, and statistical comparison to fully justify “significant up-regulation.” Also, increased gene expression does not necessarily correlate directly with photosynthetic activity.
Lines 33–34 and 36–37: Statements regarding haplotype analysis and potential genetic targets should include caveats that functional validation is necessary to confirm their effects on photosynthetic efficiency.
Lines 54–56: The notion of optimizing photosynthetic efficiency under abiotic stress should include limitations due to physiological and environmental constraints.
Lines 58–61: The sentence structure is incorrect; it should read: “Specifically, leaf morphological traits—including leaf area, petiole angle, and chlorophyll content—play a pivotal role in optimizing light capture and carbon assimilation, significantly influencing photosynthetic performance.”
Lines 70–75: Redundant sentences discussing the link between photosynthesis and oil accumulation should be merged to improve clarity and readability.
Lines 77–81: While the inefficiency of Rubisco is accurately described, additional context such as CO₂ concentration, temperature, and pH modulation should be mentioned.
Lines 85–86: The bifunctionality of Rubisco depends on the CO₂/O₂ ratio and mechanisms limiting photorespiration; this should be noted.
Lines 87–88: RbcS’s role should be described as stabilizing and regulating Rubisco rather than contributing directly to catalysis.
Lines 80–81: Correct typographical errors (“photorespiration-a process” → “photorespiration—a process”) and remove redundant phrases in lines 77–78.
Lines 101–102: Gene diversity encompasses copy number variation and regulatory sequence differences affecting expression and stress response; this should be added.
Line 105–106: Use scientific names consistently (e.g., Oryza sativa L. instead of Rice).
Lines 108–112: Replace “establishes a theoretical foundation for optimizing Rubisco activity” with “provides a theoretical foundation for identifying potential targets to optimize Rubisco activity.”
Lines 113–115: Clarify that improvements in photosynthetic efficiency through gene editing are potential, not guaranteed.
Lines 140–141: “Significant conservation” mainly refers to coding sequences; variation exists among isoforms.
Lines 141–142: Specify that the four subgroups were defined using phylogenetic methods and protein sequence analysis.
Lines 145–147: Retention of RbcS genes after polyploidization may involve neo- or subfunctionalization; this nuance should be included.
Lines 150–151: Expansion of the RbcS family is due to both segmental duplication and polyploidization, not solely chromosome doubling.
Lines 153–157: Correct typographical errors (“To analyses” → “To analyze”) and incomplete sentences: “…which serve as distinctive features for each gene and may provide clues for their functional roles.” Unique motifs suggest functional differences but require experimental validation.
Lines 158–160: Add “in most studied cultivars” to acknowledge potential variation among isoforms.
Lines 166–168: Gene counts may vary with genome version and annotation method.
Lines 170–172: Rephrase to: “…suggests that the relative positions of RbcS genes in B. napus have been largely conserved compared to its diploid ancestors.”
Lines 175–184: Segmental duplication is not the sole mechanism; tandem duplication may also contribute. Definitions of collinearity should be clarified.
Lines 241–242: Specify the exact developmental stage rather than “fertility stage.”
Line 243: Stylistic: “Heatmap analysis revealed distinct expression patterns for each gene in the family.”
Lines 245–246: Clarify potential functional redundancy as a hypothesis, not a conclusion.
Lines 248–253: Separate sentences and correct grammar: “The BnRbcS genes are crucial in photosynthesis, and we exposed 16 BnRbcS family genes to light treatment…”
Lines 255–257: Correct punctuation and spacing (“12hours” → “12 hours”). Reconsider classification as long- or short-day responsive; further experiments under different photoperiods are required.
Lines 261–262: More precise: “may belong to light-inducible promoters.”
Lines 264–271: Use consistent gene nomenclature (BnRbcS). Emphasize that functional validation is needed to confirm the relationship between haplotypes and photosynthetic efficiency.
Of course, figures, charts, and tables should be inserted at the point where they are first cited in the text.
This manuscript provides a thorough and detailed analysis of the BnRbcS gene family, including phylogeny, gene structure, chromosomal distribution, expression profiling, light responsiveness, and haplotype analysis. It offers valuable insights for understanding photosynthetic regulation and potential genetic targets in rapeseed breeding. With the recommended clarifications regarding scientific precision, experimental limitations, typographical corrections, and sentence structure improvements, the manuscript will be significantly strengthened and provide a robust reference for future studies in crop photosynthesis and breeding.
Author Response
Comments 1:Lines 21–23: The statement “enhancing photosynthetic efficiency represents a key approach for improving crop yield and biomass” is generally correct. However, it should be clarified that increased photosynthetic efficiency may lead to higher biomass, but its impact on grain or economic yield depends on carbon allocation between plant organs and is not always direct.
Response 1:We sincerely thank the reviewer for raising this valuable point, and agree with the reviewer's important comment regarding the complex link between photosynthetic efficiency and yield. We have revised the text (Lines 21-23) to clarify that while efficiency boosts biomass, economic yield is strongly influenced by carbon partitioning to harvestable organs.
Comments 2:Lines 22–23: The description of RbcS as encoding “core components of Rubisco” is misleading. RbcS does not catalyze CO₂ fixation; the catalytic activity resides in RbcL. It would be more accurate to state that RbcS “stabilizes and regulates Rubisco assembly and activity.”
Response 2:We thank the reviewer for this precise and important correction. We fully agree and have revised the sentence on Lines 22–23 accordingly to accurately reflect the structural and regulatory role of RbcS in the Rubisco holoenzyme assembly and function (Lines 24-25 ).
Comments 3:Lines 24–26: There are punctuation errors in the species names (e.g., “Brassica. napus” should be “Brassica napus L.”).
Response 3:Thank you for pointing out the irregularities in the species naming format in the manuscript. Correct academic naming is crucial. Based on your feedback and in order to improve the conciseness and readability of the manuscript, we have uniformly changed the complete Latin scientific names of all species to internationally recognized standard abbreviations that comply with botanical naming conventions in this revision.
The specific modification example is as follows:
Before modification: . across Brassica.napus, Brassica.juncea, Brassica.carinata... (with redundant punctuation)
After modification: .across B. napus, B. juncea, B. carinata...
Comments 4:Lines 27–29: The phrase “functional significance” should be tempered; bioinformatic and expression analyses suggest “potential functional roles” rather than proven significance.
Response 4:We thank the reviewer for this precise suggestion to improve the accuracy of our wording. As suggested, we have revised the phrase to “revealing the potential functional roles and regulatory complexity” to more accurately reflect the inferential nature of our bioinformatic and expression analyses ((Line 30 ).)
Comments 5:Lines 30–32: Light response experiments require details of light intensity, control conditions, and statistical comparison to fully justify “significant up-regulation.” Also, increased gene expression does not necessarily correlate directly with photosynthetic activity.
Response 5:We thank the reviewer for these critical suggestions to improve the rigor of our description. In the revised manuscript ( Methods section lines 484-485), we have now provided the detailed light intensity used, clarified the control conditions, and added the statistical comparison methods and p-values to justify the claim of “significant up-regulation.”(Figure 9) Furthermore, we have moderated the language to clarify that the observed up-regulation indicates a suggesting that their expression is enhanced under prolonged light exposure within this timeframe under 6-12hours.
Comments 6:Lines 33–34 and 36–37: Statements regarding haplotype analysis and potential genetic targets should include caveats that functional validation is necessary to confirm their effects on photosynthetic efficiency.
Response 6:We thank the reviewer for raising this important point. We have revised the statements regarding haplotype analysis and genetic targets throughout the manuscript (including on Lines 33–36) to explicitly incorporate the necessary Reminders emphasizing that the identified targets are promising candidates that require future functional validation to confirm their effects.
Comments 7:Lines 54–56: The notion of optimizing photosynthetic efficiency under abiotic stress should include limitations due to physiological and environmental constraints.
Response 7:We thank the reviewer for this important insight. We have revised the sentence to explicitly acknowledge that optimization of photosynthesis under stress occurs within, and is limited by, fundamental physiological and environmental constraints (Lines 55-60)
Comments 8:Lines 58–61: The sentence structure is incorrect; it should read: “Specifically, leaf morphological traits—including leaf area, petiole angle, and chlorophyll content—play a pivotal role in optimizing light capture and carbon assimilation, significantly influencing photosynthetic performance.”
Response 8:Thank you for your suggestion. It has been revised to: Specifically, leaf morphological traits—including leaf area, petiole angle, and chlorophyll content—play a pivotal role in optimizing light capture and carbon assimilation, significantly influencing photosynthetic performance. (Lines 60-62)
Comments 9:Lines 70–75: Redundant sentences discussing the link between photosynthesis and oil accumulation should be merged to improve clarity and readability.
Response 9:We thank the reviewer for this suggestion to improve conciseness. We have merged the redundant sentences into a single, clearer statement as follows:Given that photosynthetic carbon assimilation provides the primary carbon source for lipid biosynthesis, the functional link between photosynthetic efficiency and oil accumulation in rapeseed seeds has become a key focus for optimizing yield (Lines 71-74).
Comments 10:Lines 77–81: While the inefficiency of Rubisco is accurately described, additional context such as CO₂ concentration, temperature, and pH modulation should be mentioned.
Response 10:We thank the reviewer for this suggestion. We have revised the paragraph on Lines 77–81 to explicitly incorporate and contextualize the roles of CO2 concentration, temperature, and pH modulation in regulating Rubisco’s efficiency and the carboxylation/oxygenation balance (Lines 75-83).
Comments 11:Lines 85–86: The bifunctionality of Rubisco depends on the CO₂/O₂ ratio and mechanisms limiting photorespiration; this should be noted.
Response 11:We thank the reviewer for highlighting this important nuance. We have revised the sentence on Lines 85–86 to explicitly note that Rubisco’s bifunctional outcome depends on the CO2/O2 ratio and that mechanisms exist to limit photorespiration. The modified content can be found on Lines 87-92.
Comments 12:Lines 87–88: RbcS’s role should be described as stabilizing and regulating Rubisco rather than contributing directly to catalysis.
Response 12:We have revised the sentence on Lines 87–88 to accurately describe the role of RbcS in stabilizing and regulating the Rubisco holoenzyme, rather than implying a direct catalytic contribution. The modified content can be found on Lines 92-95.
Comments 13:Lines 80–81: Correct typographical errors (“photorespiration-a process” → “photorespiration—a process”) and remove redundant phrases in lines 77–78.
Response 13:Thank you for the reviewer's detailed corrections. We have made revisions to lines 80-81 and 77-78, and the revised content is on lines 75-83
Comments 14:Lines 101–102: Gene diversity encompasses copy number variation and regulatory sequence differences affecting expression and stress response; this should be added.
Response 14:We thank the reviewer for this insightful suggestion. We have revised the sentence on Lines 101–102 to specify that the diversity of the RbcS gene family encompasses copy number variation and regulatory sequence differences affecting expression and stress response. The revised content is on lines 75-83
Comments 15:Line 105–106: Use scientific names consistently (e.g., Oryza sativa L. instead of Rice).
Response 15:As suggested, we have replaced the common name “Rice” with the scientific name “Oryza sativa L.” in Lines 114–115.
Comments 16:Lines 108–112: Replace “establishes a theoretical foundation for optimizing Rubisco activity” with “provides a theoretical foundation for identifying potential targets to optimize Rubisco activity.”
Response 16:Thanks,The text in Lines 121–122 has been updated to the suggested phrasing.
Comments 17:Lines 113–115: Clarify that improvements in photosynthetic efficiency through gene editing are potential, not guaranteed.
Response 17:We thank the reviewer for this important clarification. We have revised the sentence on Lines 113–115 to temper the conclusion, emphasizing that improvements through gene editing are a potential outcome rather than a guaranteed result. The revised content is on lines 121-125.
Comments 18:Lines 140–141: “Significant conservation” mainly refers to coding sequences; variation exists among isoforms.
Response 18:We thank the reviewer for this precise correction. We have revised the sentence on Lines 140–141 to clarify that the “significant conservation” primarily refers to the coding sequences, while acknowledging the variation that exists among different isoforms. The revised content is on lines 163-166.
Comments 19:Lines 141–142: Specify that the four subgroups were defined using phylogenetic methods and protein sequence analysis.
Response 19:We thank the reviewer for this suggestion to clarify the methodological basis. We have revised the sentence on Lines 141–142 to specify that the four subgroups were defined using phylogenetic methods and protein sequence analysis. The revised content is on lines 165-166.
Comments 20:Lines 145–147: Retention of RbcS genes after polyploidization may involve neo- or subfunctionalization; this nuance should be included.
Response 20:We thank the reviewer for highlighting this important evolutionary nuance. We have revised the sentence on Lines 145–147 to include the possibility that retention of RbcS genes after polyploidization may involve neo- or subfunctionalization. The revised content is on lines 171-173.
Comments 21:Lines 150–151: Expansion of the RbcS family is due to both segmental duplication and polyploidization, not solely chromosome doubling.
Response 21:We thank the reviewer for this important correction. We have revised the sentence on Lines 150–151 to clarify that the expansion of the RbcS family is due to both segmental duplication and polyploidization, not solely chromosome doubling. The revised content is on lines 177-178.
Comments 22:Lines 153–157: Correct typographical errors (“To analyses” → “To analyze”) and incomplete sentences: “…which serve as distinctive features for each gene and may provide clues for their functional roles.” Unique motifs suggest functional differences but require experimental validation.
Response 22:We thank the reviewer for these detailed corrections. We have revised the sentences on Lines 153–157 to: (1) correct the typographical error (“To analyze”), (2) complete the sentence structure and improve readability, and (3) importantly, clarify that unique motifs suggest functional differences but require experimental validation, as suggested. The revised content is on lines 189-197.
Comments 23:Lines 158–160: Add “in most studied cultivars” to acknowledge potential variation among isoforms.
Response 23:The phrase ‘in most studied cultivars’ has been added to Line 196.
Comments 24:Lines 166–168: Gene counts may vary with genome version and annotation method.
Response 24:We thank the reviewer for raising this important technical point. We have added a note in the manuscript acknowledging that gene counts may vary depending on the genome assembly version and annotation methodology used. The revised content is on lines 210-212.
Comments 25:Lines 170–172: Rephrase to: “…suggests that the relative positions of RbcS genes in B. napus have been largely conserved compared to its diploid ancestors.”
Response 25:We thank the reviewer for this important suggestion. The revised content is on lines 212-213.
Comments 26:Lines 175–184: Segmental duplication is not the sole mechanism; tandem duplication may also contribute. Definitions of collinearity should be clarified.
Response 26:We thank the reviewer for these critical clarifications. We have revised the paragraph on Lines 175–184 to: (1) specify that gene family expansion involves both segmental and tandem duplication, (2) provide a clear definition of collinearity analysis and its purpose in inferring duplication history, and (3) streamline the presentation of collinearity results. The revised content is on lines 219-231.
Comments 27:Lines 241–242: Specify the exact developmental stage rather than “fertility stage.”
Response 27:We thank the reviewer for this precise suggestion. We have revised the sentence on Lines 241–242 to specify that the RNA-seq data used for expression analysis were obtained throughout the entire reproductive development period, rather than using the vague term “early stage”. The revised content is on line 309.
Comments 28:Line 243: Stylistic: “Heatmap analysis revealed distinct expression patterns for each gene in the family.”
Response 28:As suggested, we have replaced the sentence with the “Heatmap analysis revealed distinct expression patterns for each gene in the family” in line 311.
Comments 29:Lines 245–246: Clarify potential functional redundancy as a hypothesis, not a conclusion.
Response 29:We thank the reviewer for highlighting this nuance. We have revised the sentence on Lines 245–246 to clarify that functional redundancy is presented as a hypothesis derived from the expression data, not a definitive conclusion. The revised content is on lines 315-316.
Comments 30:Lines 248–253: Separate sentences and correct grammar: “The BnRbcS genes are crucial in photosynthesis, and we exposed 16 BnRbcS family genes to light treatment…”
Response 30:As suggested, We split the sentence and corrected the grammar errors. The revised content is on lines 319-321.
Comments 31:Lines 255–257: Correct punctuation and spacing (“12hours” → “12 hours”). Reconsider classification as long- or short-day responsive; further experiments under different photoperiods are required.
Response 31:We thank the reviewer for the correction and important nuance. We have corrected the spacing (“12 hours”) and, as suggested, removed the premature classification of genes as light-insensitive, noting instead the observed expression pattern between the two time points. The revised content is on lines 321-327.
Comments 32:Lines 261–262: More precise: “may belong to light-inducible promoters.”
Response 32:Thank you for the suggestion. We have revised the phrasingto the more precise statement: “may belong to light-inducible promoters.” The revised content is on lines 332-333.
Comments 33:Lines 264–271: Use consistent gene nomenclature (BnRbcS). Emphasize that functional validation is needed to confirm the relationship between haplotypes and photosynthetic efficiency.
Response 33:We thank the reviewer for these important suggestions. (1) we have standardized all gene references to the consistent “RbcS+chr+No.”(RbcSC4-1:BnaC04G0380700ZS and RbcSA4-1:BnaA04G0101200ZS) nomenclature. (2) we have emphasized that the association between specific haplotypes and photosynthetic efficiency remains preliminary and requires future functional validation to be conclusively established.
Comments 34:Of course, figures, charts, and tables should be inserted at the point where they are first cited in the text.
Response 34:All figures and tables have been placed at their first citation point, Thank you again for your suggestion.
Reviewer 3 Report
Comments and Suggestions for Authors
This study conducts multi-dimensional analyses of the RbcS genes in Brassica species, revealing that most BnRbcS genes in Brassica napus exhibit significant light responsiveness, thereby providing a theoretical basis and genetic targets for high-photosynthetic efficiency breeding in rapeseed.
However, there are also the following issues that need to be further improved.
1、Spelling errors exist in the "subcellular localization" column of some content (e.g., Table 3), where "chloroplas" should be corrected to "chloroplast".
2、Format inconsistencies in references: The reference section shows non-uniform formatting, such as missing DOI numbers for some references, incorrect publication year labels (e.g., multiple occurrences of "undefined"), and non-standard abbreviations of authors' names and journal titles, which do not meet the formatting requirements for references in academic papers.
3、Incomplete description of experimental methods: In the section "4.9 qRT-PCR Analysis under 6 hours and 12 hours light Treatments", the specific conditions of light treatment (e.g., light intensity, temperature, humidity, and other environmental parameters) are not clearly stated, nor is the selection and use of internal reference genes mentioned, resulting in reduced reproducibility of the experiment.
4、Insufficient verification of gene function: This study mainly infers the function of BnRbcS genes through bioinformatics analysis and expression pattern analysis, but lacks in vivo and in vitro functional verification experiments (such as gene silencing, construction of overexpressed plants, and determination of photosynthetic efficiency-related indicators). It is unable to directly prove the regulatory role of BnRbcS genes in the photosynthetic efficiency of rapeseed, and the persuasiveness of the research conclusions needs to be enhanced.
5、Insufficient depth in evolutionary analysis: Although phylogenetic and collinearity analyses were performed, transcriptomic or epigenetic data were not integrated to explore the evolutionary drivers of the RbcS gene family (such as natural selection, epigenetic modifications, etc.), nor were differences in the evolutionary rates of RbcS genes among different Brassica species compared. The understanding of the evolutionary mechanism of the gene family is not in-depth enough.
6、The effectiveness of data visualization needs to be optimized: The color matching and legend labeling of some figures (e.g., heatmaps, cis-acting element analysis diagrams) are not clear enough, and the presentation of some data is relatively simple (e.g., only describing the significance of differences in words). More intuitive visualization methods such as boxplots and line graphs can be added to better display the experimental results. In particular, the last subfigure of Figure 9 appears scattered and disorganized.
Author Response
Comments 1:Spelling errors exist in the "subcellular localization" column of some content (e.g., Table 3), where "chloroplas" should be corrected to "chloroplast".
Response 1:Thank you for your reminder,the spelling “chloroplas” in Table 3 has been corrected to “chloroplast”.
Comments 2:Format inconsistencies in references: The reference section shows non-uniform formatting, such as missing DOI numbers for some references, incorrect publication year labels (e.g., multiple occurrences of "undefined"), and non-standard abbreviations of authors' names and journal titles, which do not meet the formatting requirements for references in academic papers.
Response 2:TWe sincerely thank the reviewer for the meticulous examination of our reference list. We apologize for the formatting inconsistencies. We have now systematically checked and revised all references to ensure uniformity: missing DOIs have been added, publication years have been verified and corrected, and the abbreviations of authors’ names and journal titles have been standardized in accordance with the journal’s style guide.
Comments 3:Incomplete description of experimental methods: In the section "4.9 qRT-PCR Analysis under 6 hours and 12 hours light Treatments", the specific conditions of light treatment (e.g., light intensity, temperature, humidity, and other environmental parameters) are not clearly stated, nor is the selection and use of internal reference genes mentioned, resulting in reduced reproducibility of the experiment.
Response 3:TWe thank the reviewer for highlighting these omissions critical for reproducibility. In the revised manuscript, we have supplemented the “qRT-PCR Analysis” section with the detailed light treatment conditions (including light intensity, temperature, and humidity) and explicitly stated the internal reference genes selected and used for data normalization. The revised content is on lines 480-481,485-486.
Comments 4:Insufficient verification of gene function: This study mainly infers the function of BnRbcS genes through bioinformatics analysis and expression pattern analysis, but lacks in vivo and in vitro functional verification experiments (such as gene silencing, construction of overexpressed plants, and determination of photosynthetic efficiency-related indicators). It is unable to directly prove the regulatory role of BnRbcS genes in the photosynthetic efficiency of rapeseed, and the persuasiveness of the research conclusions needs to be enhanced.
Response 4:TWe sincerely thank the reviewer for this insightful comment, which accurately highlights a key limitation of our current study. We fully agree that bioinformatics and expression analyses, while strongly suggestive, are insufficient to directly prove the regulatory role of BnRbcS genes in photosynthetic efficiency. The primary aim of this work was to systematicallyidentify and characterize the BnRbcS gene family, thereby providing a foundational screen and theoretical basis for selecting prime targets for future functional studies. We have now explicitly stated in the Discussion/Conclusion that definitive proof of function requires further validation through experiments such as gene silencing, overexpression, and direct measurement of photosynthetic parameters in transgenic plants, and that this constitutes an essential direction for our subsequent research.
Comments 5:Insufficient depth in evolutionary analysis: Although phylogenetic and collinearity analyses were performed, transcriptomic or epigenetic data were not integrated to explore the evolutionary drivers of the RbcS gene family (such as natural selection, epigenetic modifications, etc.), nor were differences in the evolutionary rates of RbcS genes among different Brassica species compared. The understanding of the evolutionary mechanism of the gene family is not in-depth enough.
Response 5:TWe are grateful for the reviewer’s constructive suggestion to enhance the evolutionary analysis. Our present work aimed to establish a comprehensive genomic structural and phylogenetic framework for the BnRbcS family.Integrating evolutionary rate comparisons and transcriptomic/epigenetic data to uncover the specific drivers (e.g., natural selection, epigenetic modifications) is indeed a logical and crucial next step. Exploring these deeper evolutionary mechanisms constitutes a key objective of our subsequent research, building upon the genomic framework established here.
Comments 6:The effectiveness of data visualization needs to be optimized: The color matching and legend labeling of some figures (e.g., heatmaps, cis-acting element analysis diagrams) are not clear enough, and the presentation of some data is relatively simple (e.g., only describing the significance of differences in words). More intuitive visualization methods such as boxplots and line graphs can be added to better display the experimental results. In particular, the last subfigure of Figure 9 appears scattered and disorganized.
Response 6:TWe sincerely thank the reviewer for these detailed and constructive suggestions to improve our data presentation.We have redrawn all the images in the manuscript to improve their clarity and neatness,For data where significance was previously described only textually, we have now added more intuitive visualizations, including box plots to better display the experimental results and statistical differences. Specifically, the layout of the final sub-figure in Figure 9 has been reorganized to present the data in a more coherent and structured manner. We believe these revisions significantly improve the effectiveness and professionalism of our data visualization.